# Complete Genome and Recombination Analysis of a Novel Porcine Reproductive and Respiratory Syndrome Virus 2 (Variant 1H.18) Identified in the Midwestern USA

**DOI:** 10.3390/v17060863

**Published:** 2025-06-18

**Authors:** Joao P. Herrera da Silva, Stephanie Rossow, Igor A. D. Paploski, Mariana Kikuti, Cesar A. Corzo, Kimberly VanderWaal

**Affiliations:** Department of Veterinary Population Medicine, College of Veterinary Medicine, University of Minnesota, St. Paul, MN 55455, USA

**Keywords:** PRRSV-2, variant 1H.18, recombination, variant emergence

## Abstract

Porcine Reproductive and Respiratory Syndrome Virus 2 (PRRSV-2) represents one of the greatest threats to global pork production. Increased incidence of a genetic variant of Porcine Reproductive and Respiratory Syndrome Virus (variant 1H.18) was recently reported in the Midwestern USA Sequence comparisons in the ORF5 region indicate that 1H.18 was closely related to both sub-lineages L1H and L1C. To expand our understanding and attempt to elucidate the origin of the 1H.18, we sequenced a near-complete genome, covering all coding regions, and investigated the occurrence of recombination events that may have contributed to the emergence of the new variant. At least six distinct recombination events were identified across the coding portion of the genome. Evidence of recombination in the ORF5 region between variants 1H.31 and 1C.3 was detected. Our results suggest a likely origin of 1H.18 driven by recombination.

## 1. Introduction

Porcine Reproductive and Respiratory Syndrome Virus (PRRSV) is responsible for one of the most devastating diseases in swine. One of its main consequences is reproductive failure and the death of piglets [1,2]. In the United States alone, PRRSV-2 accounts for losses estimated at USD 1.2 billion per year [3]. PRRSV is a cosmopolitan pathogen and is present in most swine-producing countries. Two species are responsible for causing Porcine Reproductive and Respiratory Syndrome (PRRS): *Betaarterivirus europensis* (PRRSV-1) and *Betaarterivirus americense* (PRRSV-2) [4]. PRRSV populations are geographically structured, with PRRSV-1 being more prevalent in Europe and PRRSV-2 predominating in the Americas and Asia [5]. PRRSV is an enveloped, positive-sense single-stranded RNA virus belonging to the *Betaarterivirus* genus within the *Arteriviridae* family [4]. It has a genome of approximately 15 kb containing 11 open reading frames. The PRRSV-2 genome primarily encodes two large open reading frames (ORF1a and ORF1b) at the 5′ end, which, through proteolytic processing, give rise to 12 nonstructural proteins (nsp1 to nsp12) involved in viral replication. In addition to these, the genome contains nine additional ORFs that encode the virus’s structural proteins (GP2, E, GP3, GP4, GP5, ORF5a, M, and N) which are essential for virion assembly and infectivity [4].

Based on phylogenetic relationships of the ORF5 gene, PRRSV has been classified below the species level into lineages, sub-lineages, and variants [6,7,8,9,10]. PRRSV-2 exhibits high substitution rates around 1.6 × 10^−3^ to 4.7 × 10^−2^ and a high frequency of recombination, resulting in a rapid evolutionary dynamic [11,12,13,14]. It is estimated that approximately 19 new variants emerge annually in the United States [6].

A variant designated 1H.18 was first detected in 2018. Between late 2023 and 2024, 1H.18 began to gain prominence, with a noticeable increase in its detection during this period [15]. However, the actual impact of 1H.18 on U.S. swine herds remains unclear [15]. Preliminary classification analyses revealed inconsistencies in the sub-lineage assignment of this variant. Despite the high genetic relatedness among the analyzed 1H.18 ORF5 sequences, suggesting they should belong to the same clade, some were classified as sub-lineage 1C, while others were assigned to 1H [15]. It has been previously shown that ORF5 sequences belonging to 1H.18 consistently formed a monophyletic clade positioned between sub-lineages L1C and L1H [15].

We hypothesize that these classification discrepancies may result from genealogical discordance caused by recombination events. To investigate the evolutionary origin of 1H.18, we sequenced the complete coding region of its genome and examined evidence of recombination in this variant. This approach offered insights into potential evolutionary events that may have played a role in the emergence of the 1H.18 variant.

## 2. Materials and Methods

### 2.1. Sequencing and Genome Assemble

To investigate the origin of the 1H.18 PRRV-2 variant, a serum sample from off-feed sows that tested positive for PRRSV-2 by reverse transcription quantitative polymerase chain reaction (RT-qPCR) was submitted for whole-genome sequencing (WGS) at the University of Minnesota Veterinary Diagnostic Laboratory using Illumina technology (San Diego, CA, USA). RNA extractions were performed using the EZ1 Virus Mini Kit v2.0 and the EZ1 Advanced XL instrument (Qiagen, Hilden, Germany). The extracted RNA was processed for library preparation using the SMARTer Stranded Total RNA-Seq Kit v2 (TaKaRa, Kusatsu, Japan). The libraries were submitted to the University of Minnesota Genomics Center for MiSeq 150bp paired-end sequencing.

The raw reads, i.e., without any prior computational processing, were subjected to quality control analyses and adapter content assessment using the FastQC program v3 [16]. Subsequently, the reads were trimmed to remove adapters and filtered to remove low-quality bases using a Phred quality score greater than 30 [17]. The genome was assembled with SPAdes genome assembler v4.2.0 using de novo assembly strategies [18]. The reads were mapped against the assembled contigs using Bowtie2 v2.5.3 to estimate coverage and depth [19]. Prior to alignment, an index was generated from the contig file using Bowtie2. The resulting SAM file was converted to BAM, sorted, and indexed using Samtools v1.22 [20]. Coverage statistics were then calculated based on the aligned reads. The alignment of reads to the assembled contigs was visually inspected using the Tablet genome viewer v1.21.02.08 [21].

### 2.2. Sequence Comparison and Classification

Contigs were subjected to an initial search against the GenBank database using Basic Local Alignment Search Tool (BLASTn v 2.15.0+), to assess which viral sequences are most closely related [22]. To verify the completeness and assess the accuracy of the assembled genomes, we used NCBI’s ORF finder to annotate the assemblies and inspect the genomic organization [23]. The predicted open reading frames (ORFs) were compared against the canonical PRRSV-2 genome structure to confirm the presence and correct order of coding regions, including ORF1a/1b and ORF2 to ORF7. We determined the pairwise nucleotide identity percentage for the complete genome as well as separately for the ORF5 region using the Sequence Demarcation Tool v1.2-SDT v1.2 [24], which computes identity values based on true pairwise alignments, excluding gaps from the calculation. Pairwise alignments were performed using MUSCLE, as implemented within the SDT software.

### 2.3. Datasets and Pre-Processing

Two distinct datasets were assembled for the subsequent analyses: one composed of complete genomes available in GenBank [23] and another consisting of ORF5 sequences retrieved from the Morrison Swine Health Monitoring Program (MSHMP) [25], a disease surveillance program that monitors over 50% of swine herds in the United States. In total, we retrieved 1635 complete genomes from GenBank. To reduce computational costs while retaining the maximum possible genetic diversity, we derived a subset from the complete genome dataset. Subsampling was performed using CD-HIT [26] with a 5% divergence threshold; that is, for clusters of sequences sharing ≥95% nucleotide similarity, only a single representative sequence was retained. From the initial dataset of 1635 complete genomes, we subsampled 164 genomes. The complete genome sequences were trimmed by removing the 5′ and 3′ untranslated regions (UTRs), and only the coding regions were used in our analyses. The same strategy was applied to the ORF5 dataset. We initially retrieved 33,000 ORF5 sequences and used CD-HIT [26] with a 7% divergence threshold, resulting in a final subset of 274 ORF5 sequences. Multiple sequence alignments were performed using the MAFFT v.7 algorithm [27], using the default parameters.

### 2.4. Recombination Analysis

Both the complete genome and ORF5 subset datasets were initially screened for evidence of recombination. Alignments were scanned for recombination using seven different methods: Rdp [28], Geneconv [29], Bootscan [30], Maximum χ^2^ [31], Chimaera [32], Siscan [33] and 3Seq [34], all implemented in Recombination Detection Program (RDP) v. 5.67 using default parameters [35]. Recombination events were considered reliable only when statistically supported by at least four different methods with *p*-values below 0.05.

### 2.5. Phylogenetic Analisys

Phylogenetic trees were reconstructed for both complete genomes and ORF5 sequences. Maximum likelihood (ML) phylogenetic trees were inferred using IQ-TREE2 v3.0.1 [36]. To construct the phylogenies, we tested different nucleotide substitution models to identify the best-fit model for each dataset. Model selection was performed using ModelFinder, implemented in IQ-TREE, and based on the Bayesian Information Criterion (BIC). Phylogenetic trees were reconstructed using the General Time Reversible model with a proportion of invariable sites and a gamma distribution with four rate categories (GTR+I+G4) substitution model, which was identified as the best-fitting model according to the BIC for both datasets. The phylogenetic analyses were performed using 1000 ultrafast bootstrap replicates. The phylogenetic trees were visualized and annotated using FigTree V1.4.4 (http://tree.bio.ed.ac.uk/software/figtree/, accessed on 15 April 2025)

## 3. Results

### 3.1. Sequence Comparisons Reveal a Close Relationship Between PRRSV-2 Variant 1H.18 and Variants 1H.31 and 1C.3

A total of 303,212 reads was obtained, with coverage of 100% in the complete coding region of the genome and an average depth of 135×. A contig of 15,102 nt in length was obtained, corresponding to the complete coding region of the genome. To confirm that the assembled contig corresponded to variant 1H.18, we performed a classification analysis using the PRRS-Loom [6] web-based variant classification model. As expected, the model assigned the sequence to variant 1H.18 based on its ORF5 sequence.

The complete coding region was analyzed using ORF Finder (NCBI) to determine the genomic organization and identify putative open reading frames (ORFs) within the viral genome. Genome annotation confirmed that our assembly preserves the canonical genomic architecture of PRRSV-2, containing all expected ORFs. This includes ORF1a/1b, which encode the polyproteins involved in viral replication, and the structural proteins encoded by ORFs 2 through 7 [4]. These findings support both the completeness and accuracy of the assembled genome.

The assembled genome was subsequently compared against the non-redundant NCBI GenBank database using BLAST-n [22]. The results showed that our genome assembly shared 93.3% nucleotide identity with sequence MN073103.1. The ORF5 gene of MN073103.1 was classified as variant 1C.3. This sequence represents the closest full-genome match to our assembled virus. The assembled genome has been deposited in GenBank under accession number PQ252345. Since PRRSV-2 variant classification is based on phylogenetic relationships derived from the ORF5 gene [6], we conducted a BLASTn search using only the ORF5 region of our 1H.18 (PQ252345) against the non-redundant NCBI GenBank database. The top hit was sequence OR293713.1, which is classified as variant H1.31, showing 92.04% nucleotide identity.

To further investigate the genetic distinctiveness of 1H.18, we compared all available ORF5 sequences assigned to this variant against the full set of PRRSV-2 ORF5 sequences in the MSHMP database. Within 1H.18, pairwise nucleotide identity ranged from 94.4% to 100%, consistent with typical intra-variant similarity, which averages around 97.5% [6]. Among all PRRSV-2 variants, 1H.31 and 1C.3 were identified as the most closely related to 1H.18. Nucleotide identity between 1H.18 and 1H.31 ranged from 90.3% to 93.8%, while identity with 1C.3 ranged from 88.2% to 92.7%, indicating moderate genetic divergence.

### 3.2. Genealogical Discordances Between Whole-Genome and ORF5 Phylogenies Suggest Evidence of Recombination in the PRRSV-2 Variant 1H.18

Pairwise identity percentages suggest that 1H.18 could potentially be an intermediate between 1C and 1H. To further investigate this relationship, we reconstructed phylogenetic trees based on whole genomes as well as on ORF5 sequences. In the whole-genome trees, 1H.18 (PQ252345) grouped within a subclade composed of sequences assigned to 1C (Figure 1A), but these 1C viruses were not clustered with the majority of 1C viruses. On the other hand, when we inferred phylogenetic trees based on ORF5, the sequences assigned to 1H.18 formed a monophyletic clade radiating from other 1H sequences (Figure 1B). This genealogical discordance between whole genomes and may provide evidence of recombination. Similar incongruences were also observed in other clades, suggesting that they too may be affected by this mechanism. However, our focus here is on understanding the origin of 1H.18.

### 3.3. Evidence of Recombination in Both Whole-Genome and ORF5 Sequences of PRRSV-2 Variant 1H.18 Suggests an Evolutionary Origin Involving Genetic Exchange

To investigate potential evidence of recombination involving variant 1H.18, we screened complete genome datasets for recombination signals. In total, six independent recombination events were identified. Of these, four were located within ORF1a. The first event involved an unknown major parental and a minor parent belonging to sub-lineage 1H (MN865563), with a breakpoint located within nsp1 (Table 1A). The second event featured a major parental that could not be classified at the sub-lineage level (ON805850) and an unknown minor parent, with a breakpoint spanning nsp1 and nsp2 (Table 1A). The third event involved a major parent from sub-lineage 1A (MW592737) and a minor parent from sub-lineage 1C (OQ924468), with the breakpoint extending from nsp2 to nsp4 (Table 1A). The fourth event involved two parental sequences from sub-lineage 1A (MW592737 as the major parent and MW592739 as the minor parent), with a breakpoint region spanning nsp4 to nsp7 (Table 1A). The fifth event occurred in the ORF1a/1b region, specifically with a breakpoint extending from nsp7 to nsp12, involving a major parent of unknown and a minor parent from sub-lineage 1A (KT257974) (Table 1A). The sixth recombination event, detected in the complete genome of 1H.18, involved a major parent from sub-lineage 1B (OP168793) and a minor parent from sub-lineage 1H (MN865552), with the breakpoint extending from ORF2 to ORF7 (Table 1A).

Notably, no breakpoint was detected within ORF5, the genomic region typically used for variant classification. A possible explanation for this could be the absence of the true parental sequences in the dataset of complete genomes. Indeed, the ORF5 sequences of the 1C.3 and 1H.31 variants were identified as the closest relatives of 1H.18. However, complete genome sequences for 1H.31 are not available. Nevertheless, to investigate the possibility of recombination within the ORF5 region, we performed a more hypothesis-driven analysis focused on 1H.18. To this end, we retrieved all available ORF5 sequences from the MSHMP database for variants 1H.18 (*n* = 62), 1C.3 (*n* = 37), and 1H.31 (*n* = 13), totaling 112 ORF5 sequences. Subsequently, we conducted recombination analyses and identified a recombination event within the ORF5 region of 1H.18, detected across all 62 sequences analyzed and supported by six distinct methods (Table 1B). This event involved a major parental sequence from 1C.3 and a minor parental sequence from 1H.31, with the breakpoint located between positions 112 and 341, thereby supporting our hypothesis that 1H.18 may have emerged through a recombination event between a variant from sub-lineage 1C and another from sub-lineage 1H.

However, it is important to note that this evidence was based on partial ORF5 sequences, as the putative parental strains do not have complete genomes available. Sequencing of the full genomes of these parental strains would be of great value for enabling a more comprehensive investigation and confirming our observations.

## 4. Discussion

Extensive occurrence of recombination in PRRSV-2 has been reported, both among wild-type lineages and between wild-type lineages and vaccine strains, and these events sometimes contribute to the emergence of new variants [12,14,37,38]. Due to the co-circulation of numerous different PRRSV-2 lineages and variants, the Midwest region of the United States may be a favorable environment for recombination. The United States Midwest region includes the Corn Belt, where there is a high volume of animal movement, which contributes to numerous PRRSV-2 introduction events from other regions [10,39]. This creates a favorable environment for mixed infections, which are essential for recombination events to occur.

Investigations into the occurrence of co-infections by different PRRSV variants are warranted, particularly at the level of the individual host. Such studies could provide critical evidence of the potential for recombination events in commercial herds, as well as insights into possible synergistic interactions between distinct viral variants during co-infection. Currently, diagnostic sequencing is routinely performed using population samples, which limits our ability to fully understand the complexity and diversity of viral populations circulating within swine herds.

Here, we report the emergence of a new PRRSV-2 1H.18 variant in the Midwest region. Our results highlight the mosaic genome structure of 1H.18, emphasizing its complex evolutionary history and demonstrating how whole-genome analysis provides a deeper understanding of the evolutionary processes underlying the emergence of novel genetic variants. However, we analyzed only a single complete genome, additional complete genomes could provide a broader perspective on whether these recombination events are widely distributed in the 1H.18 population or restricted to the single genome we analyzed. This could offer valuable insights into the contribution of recombination to the expansion of 1H.18.

The actual impact of the emergence of 1H.18 on the swine industry remains unclear and warrants further investigation to better assess its epidemic potential [15]. Key aspects such as transmissibility, pathogenicity, and long-distance dispersal should be thoroughly explored. These insights are essential tools for informing surveillance strategies, guiding vaccine development and updates, and implementing effective control measures to mitigate potential economic losses.

## Figures and Tables

**Figure 1 viruses-17-00863-f001:**
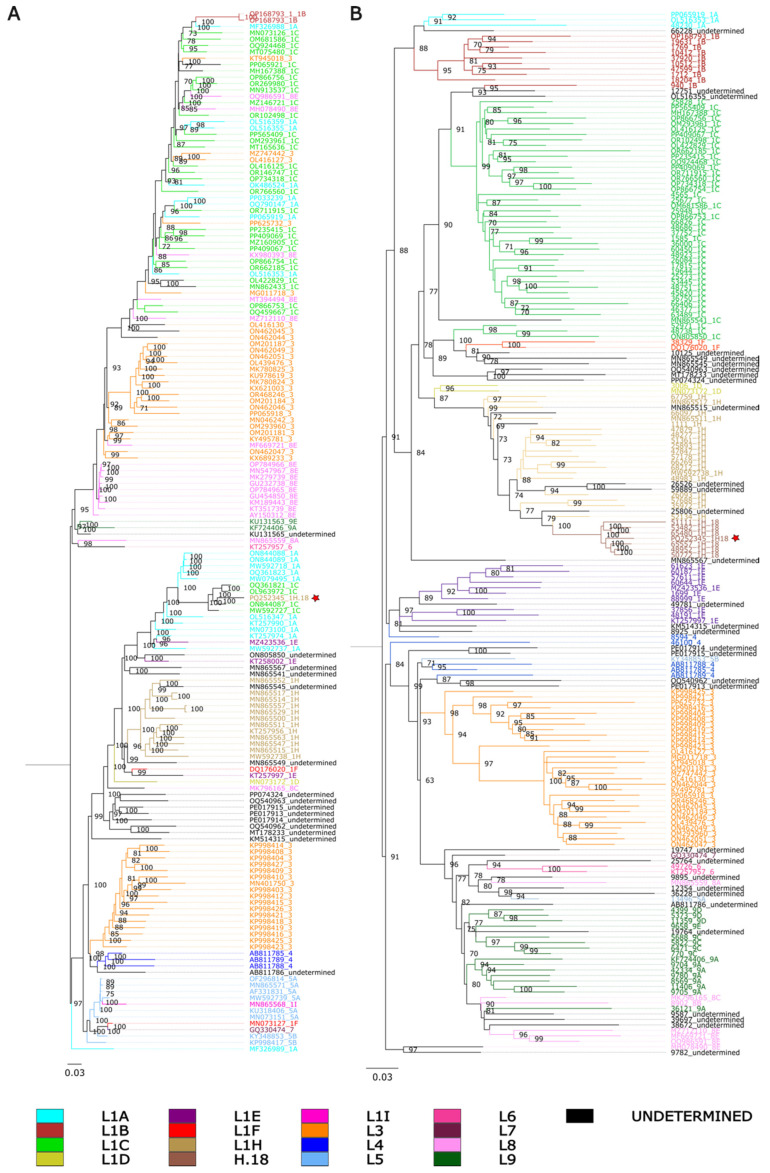
Phylogenetic tree constructed by maximum likelihood using IQ-TREE2, with 1000 bootstrap replicates and the GTR+I+G4 substitution model. Colors indicate ORF5-based lineage assignments determined by PRRSLoom [6]. Isolate PQ252345 is indicated by a red star. (**A**) Whole-genome phylogenetic tree. (**B**) ORF5 phylogenetic tree.

**Table 1 viruses-17-00863-t001:** Summary of recombination events detected in the complete genome dataset. Only recombination events detected by at least 4 different methods and with *p*-values lower than a Bonferroni-corrected α = 0.05 were considered significant. Sub-lineages of the major and minor parents are shown in parentheses.

A	Breakpoint Positions in Whole Genome								
	In Alignment		In Recombinant Sequence						
Event	Begin	End	Begin	End	Recombinant Sequence(s)	Minor Parental Sequence(s)	Major Parental Sequence(s)	Detection Methods #	*p*-Value *
1	164	476	164	473	PQ252345	MN865563 (1H)	Unknown	RBMC**S**	9.97 × 10^−11^
2	569	3224	569	2740	PQ252345	Unknown	ON805850 (Undetermined)	**R**GBMCS3	4.33 × 10^−25^
3	4246	5834	3557	5140	PQ252345	OQ924468 (L1C)	MW592737 (L1A)	RG**B**MCS3	8.41 × 10^−50^
4	5835	6952	5141	6258	PQ252345	MW592739 (L1A)	MW592737 (L1A)	RG**B**MCS3	6.02 × 10^−49^
5	6967	11,841	6363	11,222	PQ252345	KT257974 (L1A)	Unknown	MCS**3**	8.82 × 10^−34^
6	12,198	15,387	11,576	14,695	PQ252345	MN865552 (L1H)	OP168793 (L1B)	RBCS**3**	4.05 × 10^−12^
**B**	**Breakpoint Positions in ORF-5**								
	In Alignment		In Recombinant Sequence						
Event	Begin	End	Begin	End	Recombinant Sequence(s)	Minor Parental Sequence(s)	Major Parental Sequence(s)	Detection Methods #	*p*-Value *
1	112	341	112	341	PQ252345	Shmp_62265 (1H.31)	Shmp65283 (1C.3)	RBMCS**3**	4.32 × 10^−4^

# R, Rdp; G, Geneconv; B, Boostcan; M, Maxichi; C, Chimaera; S, Siscan; 3, 3Seq. * Lowest *p*-value reported by the method in bold and underlined.

## Data Availability

Portions of the dataset are privately owned by the production systems and may be subject to restrictions. However, the data can be made available upon reasonable request to the corresponding author, contingent upon permission from the respective production systems. Publicly available sequences from GenBank can be accessed and downloaded at the following link: https://github.com/herrerasilva-evol-viralland/PRRSV_L1H18 (accessed on 15 April 2025).

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
