# Peer review of "Complete Genome and Recombination Analysis of a Novel Porcine Reproductive and Respiratory Syndrome Virus 2 (Variant 1H.18) Identified in the Midwestern USA"

_viruses, 2025, doi:10.3390/v17060863_

Round 1

Reviewer 1 Report

Comments and Suggestions for Authors

The article utilizes whole-genome sequencing and recombination analysis to elucidate the recombinant origin of the novel PRRSV-2 variant 1H.18. The data are well-organized, and the conclusions are clearly articulated, providing new insights into viral evolutionary mechanisms. However, additional key details are necessary to enhance the rigor of these conclusions. Minor revisions are recommended prior to acceptance, as outlined below:

  1. Line 25 contains a spelling error in the virus species name; "Betaatervirus americense" should be "Betaarterivirus americense."
  2. Line 27 appears to have a spelling error in "Betaartevirus genus (Arteviridae family)" should be Arteriviridae.
  3. Abbreviations must be defined upon their first occurrence (e.g., "WGS" should be specified as "whole-genome sequencing").
  4. As only a single genome was analyzed, the preliminary nature of the conclusions should be clarified in the discussion. It is suggested to add: "Future studies should increase sample size to verify the prevalence of recombination events."

Author Response

We would like to thank the reviewers for taking the time to review this manuscript. Their feedback has been valuable in improving this manuscript.  Below, I have copied the reviewers’ comments in full, and a point by point response is given in Red.

Reviewer 1

Comments 1:

Line 25 contains a spelling error in the virus species name; "Betaatervirus americense" should be "Betaarterivirus americense."

Response 1:

Thank you for paying attention to this detail. However, we have double-checked the spelling on the ICTV website, and it appears to be correct.

You can verify this directly at the following link:
https://ictv.global/report/chapter/arteriviridae/arteriviridae/betaarterivirus

Please let us know if you have any further questions.

Comments 2:

Line 27 appears to have a spelling error in "Betaartevirus genus (Arteviridae family)" should be Arteriviridae.

Response 2:

Thank you very much for pointing that out. We reviewed the spelling and have corrected it accordingly.

You can verify the modification in lines 28 to 30:

"PRRSV is an enveloped, positive-sense single-stranded RNA virus belonging to the Betaarterivirus genus within the Arteriviridae family."

Please let us know if anything else needs adjustment.

Comments 3:

Abbreviations must be defined upon their first occurrence (e.g., "WGS" should be specified as "whole-genome sequencing").

Response 3:  

Excellent observation we truly appreciate your attention to detail.
We have implemented the suggested changes, which can be found in the most recent version of the manuscript, specifically on line 59:

"was submitted for whole-genome sequencing (WGS) at the University"

Please don’t hesitate to let us know if there’s anything else to review.

Comments 4:

As only a single genome was analyzed, the preliminary nature of the conclusions should be clarified in the discussion. It is suggested to add: "Future studies should increase sample size to verify the prevalence of recombination events."

Response 4:

Thank you very much for your thoughtful suggestion — it indeed highlights important gaps that warrant further investigation. We have incorporated your recommendation into the final paragraph of the conclusion. You can find the revised excerpt below:

"Here, we report the emergence of a new PRRSV-2 1H.18 variant in the Midwest region. Our results highlight the mosaic genome structure of 1H.18, emphasizing its complex evolutionary history and demonstrating how whole-genome analysis provides a deeper understanding of the evolutionary processes underlying the emergence of novel genetic variants. However, as we analyzed only a single complete genome, additional sequences are needed to determine whether these recombination events are widespread within the 1H.18 population or unique to this genome. This could offer valuable insights into the role of recombination in the expansion of 1H.18."

Please let us know if any further clarification or adjustment is needed.

Reviewer 2 Report

Comments and Suggestions for Authors

This brief report summarises the analysis of the whole genome sequence of a PRRSV-2 strain from the midwestern U.S.A.

Examination of the lineages designations and comparisons to other whole genome sequences revealed the likelihood of a sequence of recombination events having taken lace to generate this virus strain.

The approach and methods are well described and suitable for this analysis. The results are presented succinctly and clearly. The discussion summarises the findings and their impact, highlighting the value of whole genome sequencing to identify such virus strains, and attempting to clarify their origin.

Author Response

Comments 1:

Examination of the lineages designations and comparisons to other whole genome sequences revealed the likelihood of a sequence of recombination events having taken lace to generate this virus strain.

The approach and methods are well described and suitable for this analysis. The results are presented succinctly and clearly. The discussion summarises the findings and their impact, highlighting the value of whole genome sequencing to identify such virus strains, and attempting to clarify their origin.

Response 1:

We sincerely appreciate your comments, as well as the time and effort you dedicated to reviewing our work.

Reviewer 3 Report

Comments and Suggestions for Authors

The author conducted whole-genome sequencing and ORF5 sequence analysis on a newly emerged PRRSV variant, and then explored the origin of this variant. The research content is innovative, and the research results are of significance for the field to understand the variation trend of PRRSV. Overall, this paper is worthy of publication in this journal. However, please address the following issues.

  1. Verify the correctness of the term Arteviridae in the Arteviridae family.
  2. 2.2 Remove the colon in the title.
  3. Please provide complete manufacturer information for several reagents, such as TaKaRa, Qiagen, and so on.
  4. There are several abbreviations that do not have their full names written out when
  5. they first appear. Please supplement them, for example, WGS.

Author Response

Comments 1:

Verify the correctness of the term Arteviridae in the Arteviridae family.

Response 1:

Thank you very much for catching that detail. We reviewed the spelling and have made the necessary correction.

You can verify the modification in lines 28 to 30:

"PRRSV is an enveloped, positive-sense single-stranded RNA virus belonging to the Betaarterivirus genus within the Arteriviridae family."

Comments 2:

2.2 Remove the colon in the title.

Response 2:

Thank you for your attention to this detail. We have removed the colon in section 2.2, and the modification can be found on line 75.

Comments 3:

Please provide complete manufacturer information for several reagents, such as TaKaRa, Qiagen, and so on.

Response 3:

Thank you for the suggestion. Since this was a commercial kit and the protocol was followed exactly as recommended by the manufacturer without any modifications we believe it is not necessary to provide additional methodological details. These can be readily accessed in the kit’s user manual.

Please let us know if you think any clarification is still warranted.

Comments 4:

There are several abbreviations that do not have their full names written out when they first appear. Please supplement them, for example, WGS.

Response 4:

Excellent observation, thank you for your care and attention during the review of our manuscript.
We have implemented the suggested changes, which can be found in the most recent version of the manuscript, specifically on line 59:

"was submitted for whole-genome sequencing (WGS) at the University"

Please don’t hesitate to let us know if there is anything else you would like us to address.